# Sustainability, the Pontifical Academy of Sciences, and the Catholic Church's Ecological Turn

**Jaime Tatay-Nieto** 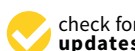

Facultad de Teología, Universidad Pontificia Comillas, 28105 Madrid, Spain; jtatay@comillas.edu;
Tel.: +34-692-664-831

**Abstract:** The promulgation of the encyclical letter *Laudato si'* by Pope Francis in 2015 has been interpreted as the final phase in the integration of sustainability concerns into Catholic Social Teaching. In this recent historical development, academic research has paid particular attention to how different eco-theological traditions, sociocultural developments, and local advocacy practices influenced the Church's ecological turn. However, the key role played by non-magisterial, intermediate institutions, particularly highly qualified interlocutors such as the Pontifical Academy of Sciences (PAS), has not been studied in depth. This article argues that, over the past 60 years, the PAS has been instrumental in this development: Raising awareness on socio-environmental issues, promoting environmental literacy, fostering ethical reflection, and catalyzing interdisciplinary dialogue in order to orient policy.

**Keywords:** environmental history; church history; greening of religion; Agenda 2030; Laudato si'; science and religion

---

## 1. Introduction

Despite having been historically marginalized from academic reflection and major international environmental fora, religion has emerged in recent decades as a relevant actor in the debate on sustainable development (Glaab and Fuchs 2018; Rakodi 2012; Deneulin and Bano 2009; Gardner 2006) and sustainability (Beling and Vanhulst 2019; Johnston 2014; Martínez de Anguita 2012). The myths, taboos, ethical codes, and cultural values conveyed by religious narratives and worldviews are increasingly being seen as valuable resources for nature conservation (Colding and Folke 2001; Berkes 1999), the transition to sustainability (Ives and Kidwell 2019), and as cultural levers with the potential to contribute to systemic social change (Otto et al. 2020; Rolston 2006).

Religious expressions, rituals, and narratives, far from disappearing, are mutating (Davie 2000; Heelas et al. 2005), resurfacing in fundamentalist forms (Almond et al. 2003), and diversifying (Berger 2014). Although the "desecularization" hypothesis (Berger 1999) and the different meanings of the term "post-secular" (Beckford 2012) are still the object of an intense academic debate, the contemporary transformation of spiritual manifestations described by sociologists of religion shows both the existence of "religioid" aspects (Benthall 2008) in the contemporary environmental movement (Harvey 1994; Orr 2003; Taylor 2009) and heightened ecological awareness and advocacy in most world religions (Gottlieb 2006; Tucker 2003), often blurring the lines between secular and faith-based organizations (Berry 2014; Johnston 2013).

Moreover, environmentalism has been described as a "revitalization movement" (Rappaport 1971; Orr 2003), a "secular faith" (Dunlap 2006), a "global faith" (Johnston 2014), and a "green faith" (Glaab and Fuchs 2018). Not only "religion has historically been a significant part of many visions of sustainability" (Johnston 2013, p. 4), but the concept of sustainability itself is also being constantly enriched and deepened by theological insights. This transdisciplinary concept articulates ecological, political, and socio-economic elements as well as cultural, ethical, and theological dimensions (Vogt and Weber 2019).

Back in the 1960s, religious anthropocentrism was perceived as a roadblock to the development of an ecological civilization (White 1967). Today, climate change skepticism is not uncommon among certain religious denominations. From a global perspective, however, religion has become an "emerging actor" (Beling and Vanhulst 2019) and an "intellectual stimulus" (Christie et al. 2019) for secular reflection in the global debate posed by Agenda 2030 (United Nations 2015). In this new context, the growing religious involvement in the environmental arena, as well as the many denominational, ecumenical, and interreligious declarations that have emanated from it (Tatay and Devitt 2017), in particular Pope Francis' *Laudato si'* encyclical letter (2015), have been received by the scientific community as a timely and necessary contribution (McNutt 2014; Schiermeier 2015; Jamieson 2015; Cafaro 2015; Raven 2016; Sánchez-Sorondo and Ramanathan 2016). However, the numerous analyses of the influential pontifical statement have mainly focused on its political implications (Pasquale 2019; Rowlands 2015), its place in the history of Catholic Social Teaching (Tatay 2018), the various theologies that converge in the document (Miller 2017), and in the influence environmental ethics and the natural sciences (Deane-Drummond 2016) had on its drafting.

In this paper, I want to highlight the central, though not sufficiently acknowledged, role that certain intermediate institutions, in particular the Pontifical Academy of Sciences (PAS), have played as hybrid forums for interdisciplinary dialogue, locus of ethical reflection, and transmission belts between magisterial pronouncements and academic research.

In order to fill this academic gap, I first focus on the recent history of the PAS in relation to the socio-environmental challenges emerging throughout the second half of the 20th and early 21st century. Secondly, I describe the main functions exercised by the PAS in order to understand its influence on the pontiffs and the key role played in catalyzing Catholic Social Teaching's "ecological turn". I argue that, within the complex institutional network of ecclesial actors and players that have facilitated the "greening" of the Catholic Church, the PAS has played a quadruple role: 1. Raising awareness on socio-environmental problems; 2. promoting environmental literacy; 3. fostering ethical reflection; and 4. catalyzing interdisciplinary research. Finally, I suggest that the ongoing science and religion dialogue taking place at the PAS since its refoundation has been instrumental in this development.

## 2. The Emerging Sustainability Challenge and the *Pontificiae Academiae Scientiarum*

The first ecclesial institution that contributed significantly to the consideration of ecological challenges in the Catholic Church was the *Pontificiae Academiae Scientiarum*. Its remote predecessor, the *Accademia dei Lincei*, was created in 1603 under the patronage of Clement VIII and was one of the first scientific academies in the West (Ladous 1994). After its restoration by Pius XI in 1936, it was transformed into an academic forum for interdisciplinary dialogue and a privileged meeting place between the Roman Catholic Church and the scientific world.

The mission of the PAS is to promote the progress of the sciences, as well as to establish a dialogue on the epistemological problems derived from interdisciplinary research (Sánchez-Sorondo 2003). The documents generated during the study weeks sponsored by the PAS are not considered magisterial teaching, but they are relevant because of the influence they have on the pontiffs, their academic rigor, and the interest they raise within the scientific community. The work of the academicians, together with the one which will be developed from 1994 by the *Pontificiae Academia Socialum Scientiarum* (PASS), the PAS sister academy, is essential for understanding the key role played by these "epistemic institutions" (Meyer 2013) in raising environmental awareness within the Catholic Church.

On the 50th anniversary of its re-establishment, G. B. Marini-Bettòlo, an Italian chemist and president of the PAS from 1988 to 1993, stated:

> "The Academy played an important role in suggesting answers to the questions presented to the Holy See by international organizations or by individual scientists, not only on the technical and scientific level, but also on the ethical and moral level. For example, on questions related to desertification, water supply, the correct use of computers, the ethics of scientific research [...]". (Marini-Bettòlo 1987, p. 79)

Moreover, "having such high-level authorities at its disposal, the Academy was in a position to make pronouncements and contributions in advanced interdisciplinary fields" (Marini-Bettòlo 1987, p. 50). This is confirmed by Peter H. Raven, former president of the American Association for the Advancement of Science and longtime member of the PAS:

> "Because of the existence of the PAS and its consistent input of objective scientific advice, the Catholic Church has accepted both biological and cosmic evolution since the 1930s and global warming since it was established as an important factor determining our common future". (Raven 2016, p. 253)

Over the past 60 years, the PAS convened study weeks, organized symposiums, and established working groups on a wide range of topics related to sustainable development and ecology (Table 1). The fruitful dialogue between the Vatican and the academicians has been the main venue of Catholic engagement with modern science. According to Christiana Z. Peppard (2015), the long history of this engagement has gone through four different eras: Astronomy and physics in the 16th to 18th centuries; geology and evolutionary theory in the 19th and early 20th centuries; bioengineering in the mid to late 20th century; and, finally, ecology and sustainability in the early 21st century.

**Table 1.** Sustainability-related study weeks, study days, symposiums, and working groups organized or sponsored by the Pontifical Academy of Sciences (1968–2015).

| Year | Topic |
| --- | --- |
| 1968 | Organic Matter and Soil Fertility |
| 1976 | Natural Products and the Protection of Plants |
| 1978 | Science and the Modern World |
| 1978 | Use of Fertilizers and its Effect in Increasing Yield with Particular Attention to Quality and Economy |
| 1980 | Mankind and Energy: Needs, Resources, Hopes |
| 1983 | Chemical Events in the Atmosphere and their Impact on the Environment |
| 1984 | Energy for Survival and Development |
| 1987 | Modern Approach to the Protection of the Environment |
| 1989 | Science for Development in a Solidarity Framework |
| 1990 | Man and his Environment. Tropical Forests and the Conservation of Species |
| 1993 | Chemical Hazards in Developing Countries |
| 1994 | Population and Resources |
| 1998 | Changing Concepts of Nature at the Turn of the Millennium |
| 1998 | Geosphere–Biosphere Interactions and Climate |
| 1999 | Science for Survival and Sustainable Development |
| 2001 | The Challenges for Science. Education for the Twenty-First Century |
| 2004 | Interactions between Global Change and Human Health |
| 2005 | Water and the Environment |
| 2010 | Nuclear Disarmament, Non-Proliferation, and Development |
| 2011 | Fate of Mountain Glaciers in the Anthropocene |
| 2013 | The Emergency of the Socially Excluded |
| 2014 | Sustainable Humanity, Sustainable Nature: Our Responsibility |
| 2015 | Climate Change and the Common Good: A Statement of the Problem and the Demand for Transformative Solutions |
| 2015 | Protect the Earth, Dignify Humanity: The Moral Dimensions of Climate Change and Sustainable Development |

As Marcelo Sánchez-Sorondo, its current president, affirms: "The Pontifical Academy of Sciences has thus become one of the favoured forums for the dialogue between the Gospel and scientific culture" (Sánchez-Sorondo 2003, p. 18). According to Ladous (1994), since the time of Pius XI, who reestablished the PAS in order to have a scientific senate parallel to the cardinal's senate, his successors have widely profited from the input of the scientists. In fact, it is in the exchange with the academicians where "the pope gets access to the scientific expertise of people at the top of their fields" (Seife 2001, p. 1472).

Yet, it should be noted that the PAS is not a research-oriented institution, but rather a policy-oriented one. The Church's global spiritual, political, and moral clout is one of the reasons why so many world-class scientists, of all religious backgrounds and none, are eager to participate, discuss, and share their expertise (Jamieson 2015; Seife 2001; Gould 1997; Singer 1991). The non-confessional character of the Academy and "the climate of mutual listening and serene encounter on subjects of great relevance" (Sánchez-Sorondo 2003, p. 20), are also pointed out as another defining trait of this unique venue. The nature of the interaction between the PAS and the official Magisterium is one of mutual influence and permanent dialogue. The Vatican has occasionally set the agenda of the PAS, but more often the academicians have freely chosen the topic of their meetings attracting the attention of the pontiffs.

By facilitating the establishment of interdisciplinary bridges and hybrid forums for dialogue, the PAS has not only informed the Holy See about various technical and scientific aspects; it has also promoted ecological literacy, influencing indirectly, but significantly, the ecclesial reception and formulation of very different socio-environmental issues such as the use of fertilizers and pesticides in agriculture (1976), the ethics of scientific research (1978), the role of fossil fuels in energy generation (1980, 1984), the destruction of the ozone layer (1983), chemical pollution (1983, 1993), the use of natural resources and environmental protection (1987, 1989), the centrality of energy for development (1980, 1984), the conservation of biodiversity in tropical forests (1990), the connection between population growth and resource depletion (1994), the loss of biodiversity (1998), science education (1998, 2001), climate change (1983, 1998, 2011, 2015), sustainable development (1999), water (2005), nuclear disarmament (2010), and social exclusion (2014, 2015).

In sum, the role of the PAS has been instrumental in attracting the Church's attention to the debates taking place in the scientific community and in offering, from an academically authoritative voice, plausible ethical and political responses to the emerging sustainability challenge. There is ample historical evidence that this has been the case.

### 3. The Historical Role of the PAS in Raising Awareness on Environmental Issues in the Catholic Church

It is not a coincidence that, since its reestablishment in 1936, successive popes have shown great interest in the work of the PAS, considering it an authorized interlocutor with whom to establish a dialogue, as well as a valuable source of knowledge with which to elaborate their speeches, exhortations, and encyclicals on a wide variety of issues. An example of this is the numerous papal interventions addressed to the academicians at Casina Pio IV, the Academy's headquarters in the Vatican (Pontifical Academy of Sciences and Pontifical Academy of Sciences 2003).

During World War II, before the new ecological conscience had emerged, Pius XII already warned, unaware of the worst horrors of that conflict, that new scientific advances could "become a double-edged sword and bring both health and death" (Pius XII 1941). After the war, in 1968, the PAS organized a study week to analyze the issue of soil fertility and, indirectly, the use of fertilizers and pesticides in industrial agriculture (Pontifical Academy of Sciences 1968, 1978b), echoing the problem that motivated Rachel Carson to write *Silent Spring* (Carson 1962), the essay that kicked-off the environmental movement.

In 1972, the year of the historic United Nations Conference on the Human Environment, the first secular president of the PAS, Brazilian biophysicist Carlos Chagas, was elected. In Marini-Bettòlo's words, this is when the Academy was reoriented to "become an active center of interaction between academics and the international scientific community, capable of facing scientific problems and applying the resulting solutions to the problems of the modern world" (Marini-Bettòlo 1987, p. 53).

During the numerous study weeks organized by the PAS, the interrelated scientific, technical, and economic dimensions of the issues previously selected by the academicians were addressed in conjunction with the political, social, cultural, and ethical implications that are also essential for facing the complexity of contemporary challenges (Pontifical Academy of Sciences 1978a). During the 1970–1990 period, the foundations of integral ecology began to be established. Problems as

diverse as extreme poverty, economic inequality, and the inefficient management of natural resources were increasingly related by the academicians to the need for environmental governance, ethical development, indigenous knowledge, and ecological literacy.

As John Paul II (1989, pp. 4–5) himself acknowledged, during his pontificate the PAS was slowly transformed into a two-way communication channel: on the one hand, it brought scientific knowledge about the environment to the Vatican; on the other hand, it became a unique place for interdisciplinary dialogue able to influence secular thinkers on ethics. In the 1990s, following the historic Earth Summit in Rio de Janeiro (1992) and the publication of an influential declaration signed by 1700 leading scientists, The World Scientists' Warning to Humanity (Union of Concerned Scientists 1991), the PAS continued to address several pressing socio-ecological issues.

For example, when discussing the transfer of polluting industries to developing countries with low capacity to treat, store, and transport toxic waste (Dardozzi and Ramel 1996; Pontifical Academy of Sciences 1994), John Paul II denounced the increased chemical risks of those exposed and the hidden environmental injustice:

> "The serious abuse and offence against human solidarity when industrial corporations in rich countries take advantage of the economic and legislative weakness of poorer countries to locate their production plants or to locate waste that will have a degrading effect on the environment and on people's health". (John Paul II 1993, p. 2)

He also remarked that the reflection emanating from the PAS is not, properly speaking, magisterial, "but it is pontifical", and "scientific analysis is precisely what the pontiffs have always asked of it and what the Academy has given them throughout its history" (John Paul II 1993, p. 3). Furthermore, as he remarked two years before in the same venue:

> "The data that emerge from your research and discussions will therefore be important and very useful in helping the Holy See to formulate and clarify—in accordance with its own mission and responsibility—appropriate guidelines and suggestions". (John Paul II 1991, p. 1)

The two issues addressed by the PAS in 1998, one of an epistemological character—the concept of nature—and the other of a global nature—the interaction between the geosphere, the biosphere, and the climate—also provide an opportunity to see the dynamic at play in the Academy (Pontifical Academy of Sciences 2000). Both meetings indirectly stimulated ethical, philosophical, and theological reflection on the anthropological metaphors that have conveyed the dominant cultural paradigm in the West and on the need to adopt a broader spatial and temporal planetary framework: "The clarification of the biogeochemical cycles of nature is one of the great scientific challenges of our time", stated the academicians; the study of these cycles "demands an understanding of the complex system of interactions that sustains life on Earth" (Bengtsson and Hammer 2001, p. xi; Marini-Bettòlo 1994). Both sets of questions, instrumental in the development of sustainability science as an integrated discipline (Odum 1977), attracted the attention of John Paul II (1991, p. 4).

Another set of issues highlighted by the PAS at the turn of the millennium, and central to the formulation of integral ecology, is the need for an interdisciplinary approach to the concept of sustainable development (Pontifical Academy of Sciences 2000; Keilis-Borok and Sánchez-Sorondo 2000) and the centrality of education in addressing the challenges of science (Pontifical Academy of Sciences 2002). For the academicians, a holistic view of education is also essential, "to be aware of the interdependence with the environment and the universe" and "to enable contributions to the solution of the acute problems facing humanity (poverty, food, energy, and environment)" (Pontifical Academy of Sciences 2002, p. 291).

Once again, John Paul II addressed the academicians to recognize "the increasing damage caused by modern civilization to people, the environment, climatic conditions and agriculture", and, noting that some are anthropogenic, argued that "it is man's responsibility to limit the risks to creation" (John

Paul II 1999, p. 2). Echoing the interest of the scientists, he also proposed "an education in human and moral values" and called for conversion: "A profound personal conversion in one's relationship with others and with nature" which "will allow for a collective conversion and a life in harmony with creation" (John Paul II 1999, p. 7).

Following the formulation of the Millennium Development Goals, the UN began the deliberation process that would lead to the Agenda 2030 (United Nations 2015) and the establishment of 17 Sustainable Development Goals (SDG). During this period, the academicians focused their interests on several challenges related to sustainability such as human health and global change (2004), environment and water (2005), nuclear proliferation (2010), and climate change (2011, 2015).

As it did before, the PAS addressed three interrelated issues—economic development, global biogeophysical change, and the implications of both for human and animal health—noting that "human activity and the global environment have become inseparable" (Pontifical Academy of Sciences 2006, p. xv). The 2005 workshop titled Water and the Environment framed the technical debate on water management within a broad socio-environmental vision that connected five related issues: Biodiversity, hydrology, climate change, land-atmosphere interactions, and watershed management (Rodríguez-Iturbe and Sánchez-Sorondo 2007).

During the pontificate of Benedict XVI, the academicians addressed again the challenge of nuclear weapons proliferation taking into account its implications for development, the economy, energy, and climate change (Pontifical Academy of Sciences 2010, p. 15). The following year, a brief report prepared by the PAS (2011) on the melting of mountain glaciers warned of the implications of this phenomenon for the populations that depend on these vital reserves for their survival. The neologism "anthropocene", coined in 2002 by academician and Nobel laureate Paul Crutzen, was used for the first time in an ecclesial document (Pontifical Academy of Sciences 2006) assuming the conclusions of the IPCC, echoing its warning call on the effects of climate change, and calling for political action.

By request of Francis, in the run-up to the promulgation of the encyclical letter *Laudato si'*, the PAS and PASS (2014, 2015) studied the problem of social exclusion linking the challenge of poverty and marginalization to climate change and energy access (Pontifical Academy of Sciences et al. 2015; Sánchez-Sorondo 2015; Pontifical Council of Justice and Peace 2014). Sachs (2015), the architect of the Agenda 2030 and director of the UN Sustainable Development Solutions Network, framed the debate in the emerging paradigm of the SDGs, stressing the need for an ethical analysis of the challenge of sustainability. Similarly, in the workshop Sustainable Humanity, Sustainable Nature (2014), the academicians from both the PAS and the PASS reflected on the paradigm of socio-environmental systems and the new, emerging conceptual framework of the SDGs. In their view, the three interrelated crises of economic poverty, social exclusion, and environment degradation will not be solved with economic instruments and technical innovation unless there is also a shared ethical commitment and coordinated political action (Pontifical Academy of Sciences and Pontifical Academy of Social Sciences 2014, p. 3; Archer and Donati 2008).

A year later, right before the promulgation of *Laudato si'*, both academies jointly addressed the climate emergency in a statement (Pontifical Academy of Sciences and Pontifical Academy of Social Sciences 2015) prepared in the run-up to COP21. The document quotes the latest IPCC and WB reports, proposing as a solution of the "intertwined crises of poverty, exclusion and environment" a "cooperative holistic strategy" based—as the final declaration of Rio + 20 (2012) suggested—on the tripod of economic progress, social inclusion, and environmental sustainability. However, compared to the Rio + 20 declaration, the scientists explicitly mentioned the need for a "moral revolution" or "attitudinal reorientation" capable of impacting at the cultural level. Along with a comprehensive package of institutional, political, technical, and financial reforms that would allow our civilization to be reoriented towards the SDGs, the scientists emphasized the role that religious traditions play in the type of cultural transformation needed to address the sustainability crisis.

This synthetic historical overview shows that, over the past 60 years, the PAS has played a quadruple role in the "greening" of the Catholic Church: Raising awareness on socio-environmental

issues, promoting environmental literacy, fostering ethical reflection, and catalyzing interdisciplinary dialogue. To a great extent, the instrumental role played by the Academy has been possible because of its peculiar institutional location and structure as well as its interest in the science and religion dialogue.

## 4. Conclusions

According to Christiana Z. Peppard, the question, for both the Vatican and the academicians, "is not whether science and religion can coexist. The question is how scientific advancement informs theological interpretation and ethical reasoning in a world of myriad mutual dependencies" (Peppard 2015, p. 37). Since its refoundation in 1936, the academicians have been concerned with studying emerging scientific and technical challenges, but also with promoting ethical reflection in order to orient policy and guide the Church's magisterium.

Given the profoundly interdisciplinary character of sustainability science (Odum 1977; Rolston 2006; Thorén and Persson 2013; Annan-Diab and Molinari 2017), the natural scientists of the PAS—and, from 1994, together with the social scientists of the PASS—have become partners in a meaningful dialogue between the natural sciences, the social sciences, philosophy, and theology. In fact, the Academy has unofficially functioned as the top-notch scientific board in the Catholic Church and, not surprisingly, the first ecclesial institution who payed careful and sustained attention to the emerging socio-ecological challenges facing humanity—transmitting the pressing ethical questions raised by the academicians to the pontiffs and, through their global reach, to politicians, religious leaders from other faith traditions and civil society as a whole. Its indirect influence cannot be downplayed.

If, as Elinor Ostrom et al. put it, "institutional diversity may be as important as biological diversity for our long-term survival" (Ostrom et al. 1999, p. 278), we could argue that the institutional inner diversity of a world religion such as the Catholic Church has analogously played a key role and may be as important as the official magisterium itself in tackling the sustainability challenge, the single greatest threat to our long-term survival.

Christopher D. Ives and Jeremy Kidwell affirm "that while there is much potential support for human values for sustainability within religious traditions, it is essential that religion is seen as a complex, multi-scalar and multi-dimensional institutional phenomena" (Ives and Kidwell 2019, p. 1355). They distinguish between the individual, the community, and the formal institutional scale. These three levels interact with one another and are permeable to the sociocultural and ecological context within which they are embedded. For Ives and Kidwell, the formal institutional scale includes "public statements by major religions or denominations", such as *Laudato si'*; the community scale represents "the teachings and viewpoints of particular churches, temples or faith communities"; and the individual scale "is the values held by individual members of these communities, which may be highly diverse and conflict at times with the values espoused at the other scales" (Ives and Kidwell 2019, p. 1358).

This three-layered scheme is useful to understand how social values, ethical concepts, and theological insights are received, reinterpreted, and transmitted within world religions at different levels. In his analysis of *Laudato si'*, the most authoritative Catholic statement on ecology to date, Kevin O'Brien concludes: "From the encyclical, Christian ecological ethicists can learn about the importance of identifying spatial and temporal scales in moral terms and the usefulness of hierarchical levels that distinguish between local, community, and global concerns" (O'Brien 2019). However, when it comes to sustainability, not only ethicists can learn about the centrality of multiscalar thinking and hierarchical organizations. In order to engage in interdisciplinary dialogue and cooperate, theologians, philosophers, historians, economists, and policymakers can learn from it too. As Nathan Schneider puts it, cooperation is "organization in depth" (Schneider 2019, p. 146).

The question of whether the principle of subsidiarity has played a major role in the institutional development of the Church and in the emergence of an interdisciplinary playing field, and a "multiscalar perspective" (O'Brien 2008) on environmental issues, remains an open one. However, there is sufficient evidence to argue that intermediate institutions such as the PAS (and also, to a lesser extent, Caritas

Internationalis, the Pontifical Academy for Life, the Pontifical Council for Justice and Peace, and the Pontifical Academy of Social Sciences) should be considered legitimate "communities of discourse" (Shiffman 2019, p. 92) within the Church's structure and, thus, relevant, non-magisterial actors in the ecclesial hierarchy.

It is also worth noting the interaction between the PAS, an "epistemic institution" based on equality and dialogue that "assimilates basic scientific and technical research and applies it to specific legal or policy problems" (Meyer 2013, p. 17), and the Vatican, a hierarchical institution based on tradition and authority that promulgates official Magisterium and ethical guidelines. The asymmetric interplay between the two institutions requires further research since it could shed light on how other actors in the environmental arena may engage in a deliberative process. If, as Johnston argues, "in the end sustainability is not a goal or end-point, but something closer to a process of community discernment, and a strategy for engaging with others who do not share the same values or vision of the future" (Johnston 2013, p. 5), then the long history of the engagement with different sciences, philosophies, and ethical traditions at the PAS becomes a paradigmatic case study in the worldwide discernment process set up by the Agenda 2030.

Collective action between religious and secular environmental movements is still rare. Information exchange is widespread between these two groups, but interorganizational ties and coordinated advocacy strategies are not the norm (Ellingson et al. 2012). In light of these difficulties, the history of the PAS shows that the very different spiritual and academic backgrounds of the academicians have not been a roadblock, but rather a catalyst for communal discernment, ethical reasoning, interdisciplinary research and, to a lesser extent, political advocacy. The lesson learnt from the historical PAS–Vatican dialogue indicates that institutional frameworks matter and can lead to fruitful communal processes of discernment and collaboration. This is a way of proceeding that could be scaled up to "strengthen the means of implementation and revitalize the global partnership for sustainable development" (SDG17).

**Funding:** This research received no external funding.

**Acknowledgments:** The author thanks Celia Deane-Drummond, Michael Schuck, and two anonymous reviewers for providing constructive comments which improved the article.

**Conflicts of Interest:** The author declares no conflict of interest.

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
