# Peer review of "Sustainability, the Pontifical Academy of Sciences, and the Catholic Church’s Ecological Turn"

_religions, doi:10.3390/rel11100488_

Round 1
Reviewer 1 Report
I have doubts about the title of the article. I believe that to achieve the goal set by the author/s, it is worth considering also the role of the Pontifical Academy of Social Sciences to a greater extent. The title in such a case could be more general and concern the contribution of all papal scientific societies to the teaching of popes on the environmental crisis and building a sustainable world.
Author Response
Thank you for your time, first of all.
Yes, I agree that expanding the scope of the article in order to consider the role of the PASS (and other Pontifical Academies/Ecclesial bodies) is highly relevant, but it would broaden the scope of the paper too much.
I will point out to the importance of these other institutions and signal the necessity for further research.
Thank you again.
Reviewer 2 Report
The author wants « to highlight the central, though not sufficiently acknowledged, role that certain “interstitial institutions”, in particular the Pontifical Academy of Sciences (PAS) , have played as hybrid forums for interdisciplinary dialogue, locus of ethical reflection, and transmission belts between magisterial pronouncements and academic research ». The article responds to this objective in a clear and concise manner. The article itself is short (8 pages), but this is sufficient to provide the necessary information. Readers interested in learning more can refer to the extensive list of references that follows the article (including PAS publications). According to me, therefore, the article can be accepted in its current form.
Author Response
Thank you for your time and willingness to review the draft.
Reviewer 3 Report
The article represents a valuable and original contribution to investigate the sources of the "ecological turn" in the Catholic Social Teaching. However, a few additional clarifications may improve this already excellent study. First, it should be made clear that the "greening" of religions discussed in the introduction is by no means a linear process. For instance, there is a fierce battle of ideas inside the American Christian field, with some positions (as in the case of the conservative Evangelicalism and to some extent even within the US Catholic Church) strongly opposed to the Christian-liberal approach to environmental issues. A debate on the so-called anthropocentrism in the creation is the cornerstone of this conflict of visions. Second, in my modest opinion the Author should clarify the nature of the interaction between the PAS and the magisterial role of the Popes, in terms of "causality", mutual influence, agenda setting for the discussions (line 230 mentions "by request of Francis"; line 270 refers to PAS as "first ecclesial institution"; the role of the PAS as "interstitial institution" in line 57 doesn't seem consistent with its function as "scientific senate parallel to the cardinal's senate" in line 114 since the PAS itself doesn't have any deliberative mandate). Third, in my opinion the Author needs to introduce the distinction/interaction between an epistemic institution such as the PAS (based on scientific knowledge and equality) and a religious hierarchical institution (based on tradition and authority) like the Catholic Church. The balance between the two becomes quite interesting to investigate when deliberative processes are involved. All in all, it is a very stimulating and sophisticated analysis of a neglected aspect of the institutional diversity inside the Vatican.
Author Response
Thank you, first of all, for your time and willingness to review this draft.
- Your point is true, though I think the US is the exception rather than the norm. From a more universal perspective, the "battle of ideas" is not as fierce as it seems to be in the Evangelical (and even Catholic) US Churches. At least in Europe, where I'm based, all mainstream churches are quite aligned on environmental issues. Yet, I will mention Lynn White's critique (which kick-started the debate on religious environmentalism) at the introduction in order to offer a more nuanced view of the Christian involvement on ecological matters.
- Your suggestion is very helpful. I clarifies the nature of the interaction between the PAS and the Vatican. I believe it has been of "mutual influence" or "bi-directional" depending on the topic and the interests/concerns of a particular Pope.
- This is also a very important distinction I need to introduce. I''l briefly build on Timothy Meyer's distinction and make explicit at the conclusive section that this is a point that requires further research.
Thank you, once again, for your attentive reading.